# Geospatial Assessment of Flood-Tolerant Rice Varieties to Guide Climate Adaptation Strategies in India

Nisha Koppa [1] and Giriraj Amarnath [2,*]

1   International Water Management Institute, New Delhi 110012, India; nishakoppaa@gmail.com
2   International Water Management Institute, Battaramulla 10120, Sri Lanka
*   Correspondence: a.giriraj@cgiar.org; Tel.: +94-11-2880000

**Abstract:** Rice is the most important food crop. With the largest rain-fed lowland area in the world, flooding is considered as the most important abiotic stress to rice production in India. With climate change, it is expected that the frequency and severity of the floods will increase over the years. These changes will have a severe impact on the rain-fed agriculture production and livelihoods of millions of farmers in the flood affected region. There are numerous flood risk adaptation and mitigation options available for rain-fed agriculture in India. Procuring, maintaining and distributing the newly developed submergence-tolerant rice variety called Swarna-Sub1 could play an important role in minimizing the effect of flood on rice production. This paper assesses the quantity and cost of a flood-tolerant rice seed variety- Swarna-Sub1, that would be required during the main cropping season of rice i.e., *kharif* at a district level for 17 major Indian states. The need for SS1 seeds for rice production was assessed by developing a geospatial framework using remote sensing to map the suitability of SS1, to help stakeholders prepare better in managing the flood risks. Results indicate that districts of Bihar, West Bengal and Uttar Pradesh will require the highest amount of SS1 seeds for flood adaptation strategies. The total estimated seed requirement for these 17 states would cost around 370 crores INR, less than 0.01 percent of Indian central government's budget allocation for agriculture sector.

**Keywords:** remote sensing; GIS; flood tolerant seeds; Swarna-Sub1 Rice; climate adaptation

## 1. Introduction

Floods are amongst the most common natural disaster across the globe. They pose a threat not only to the environment but also to society as they endanger lives, properties and livelihoods of the people. The report and analysis collected by UNISDR and the Belgian-based Centre for Research on the Epidemiology of Disasters (CRED) highlighted that in the year 2021, the impacts of floods were felt heavily across the developing countries in Africa and Asia [1]. For populous continent such as Asia, around 42% of the global flood events occurred between 1950–2020, which affected around 3.65 billion people and economic losses accounted to USD 556 billion [1,2]. Within Asia, the south Asian region is highly vulnerable to flood impacts. Recent estimates for the South Asian region shows that, between 2000 and 2020, these countries have experienced 11% of the world's natural disasters and 12% of floods and droughts, making over 700 million people and 190 million ha of agricultural land vulnerable [1]. Considering the increasing global temperature, unplanned urban growth and environmental degradation, it is likely that the frequency and severity of flood risks will increase in the exposed countries such as Bangladesh, India and Nepal [3]. Additionally, for largely agrarian countries such as India and Bangladesh, these changes especially threaten the agriculture sector, as it increases the ambiguity for the small-scale and poor farmers whose livelihoods are dependent on the agricultural production in these regions.

For a large and populous country like India, the increasing weather variability and the subsequent impact of disasters such as floods is concerning. For instance, official

statistics reveal that 15% of the total area in India (which amounts to approximately 49.82 million hectares) is extremely vulnerable to floods [4]. Moreover, the variable summer monsoon in India has often precipitated floods, especially in the basins of the Himalayan rivers. These large river basins, such as the Indus, Ganges and Brahmaputra, cause significant monsoon runoff, leading to immense flooding in the plains [5]. Considering that these rich and fertile plains are used for agriculture production, frequent floods in the region affects the people dependent on agriculture. One of the most commonly grown crops in the fertile plains of Indo-Gangetic River basin is 'Rice' (also referred to as paddy in this study). Currently, rice is grown across 43.86 million hectare of area and the production level is 104.80 million tonnes in India [6]. The rice crop which requires a lot of water, is commonly sown during the months of July-October i.e., during the monsoon season in India. The rice farmers in this region take a heavy toll as the recurring floods just after crop sowing leads to crop losses. While rice crop can thrive well in flooded soils, the crop is still vulnerable to complete submergence for longer days and around 16% of the world's rice production area is affected by recurring submergence due to flash floods [7–9]. These recurring impacts of floods in India necessitate improving the farmer's knowledge with regards to adapting and coping methods along with improving flood-resilient infrastructure to reduce the damaging impacts on farming communities.

## 1.1. Extent and Impact of Flooding on Rice Production

Rice production in rain-fed low lands is often severely affected as the crop at different growing stages suffers from various stresses, such as limited gas diffusion, effusion of soil nutrients, mechanical damage, increased susceptibility to pests and diseases and stresses due to low-light due to flooding (also called as submergence) or water-logging [10,11]. The frequent flooding during rice cultivation (which occurs during the monsoon months) in rain-fed lowland areas of South Asia leads to a complete submergence of the rice crop for approximately 10–15 days. While rice has some adaptive traits for tolerance to submergence, the low-land rice cultivars used in South Asian countries are still sensitive to complete submergence [12]. In India, the Indo-Gangetic River basin, which is a favourable belt for rice cultivation, is also the most flood-affected region in India. Moreover, around 30% of the total rice growing area, which amounts to 12–14 million ha is prone to flash flooding with an average productivity of only 0.5–0.8 tonnes per ha as compared to 2 tonnes per ha in favourable lowlands [13]. With a high incidence and severity of floods, small and poor farmers incur heavy economic losses. Figure 1 shows the average agriculture area (in hectare) affected by floods in 17 major Indian States. It can be seen from Figure 1 that Bihar is the worst flood hit state in India followed by Uttar Pradesh, West Bengal and Assam- all situated in the Indo-Gangetic plains, i.e., each has one of the two major rivers (or its tributaries) of Ganga and Brahmaputra rivers flowing from the Himalayas. In Bihar, nearly 73% of the total geographical area and 76% of its total population are constantly under the threat of flood [14]. Almost every year, there is severe flooding in the state of Bihar which causes loss to lives, properties and livelihood [15]. With the onset of the monsoon, the rivers originating from Himalayas flow down with massive force, causing rivers such as *Koshi* and *Ganges* to rise above the danger level, which leads to severe floods in northern parts of Bihar.

In Figure 2, the graph plots the state-wise five-year average rice production in India. It can be seen that West Bengal produces around 15,000 tonnes of rice on average per year, followed by Uttar Pradesh, Punjab, Andhra Pradesh and Odisha. Comparing this with the agricultural flooding, it can be seen that states which experience high levels of flooding are also amongst the top producers of rice, except Bihar. For instance, the eastern state of West Bengal ranks first in rice area and production in the country. However, around 30% of the rice growing area in this state comes under the rain-fed lowlands which suffer from frequent flash floods due to unpredictable rainfall during the major rice growing season (*kharif*), leading to a drastic reduction in yield [13,16].

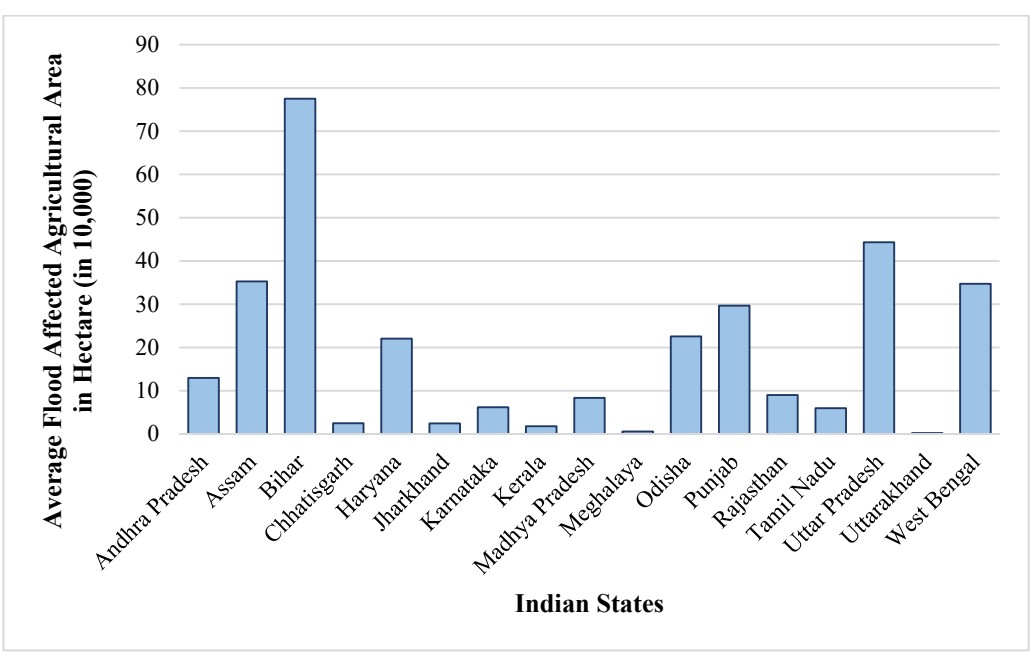

**Figure 1.** Average flood affected agricultural areas between 2000 to 2018 for the 17 Indian States in India. Source: IWMI.

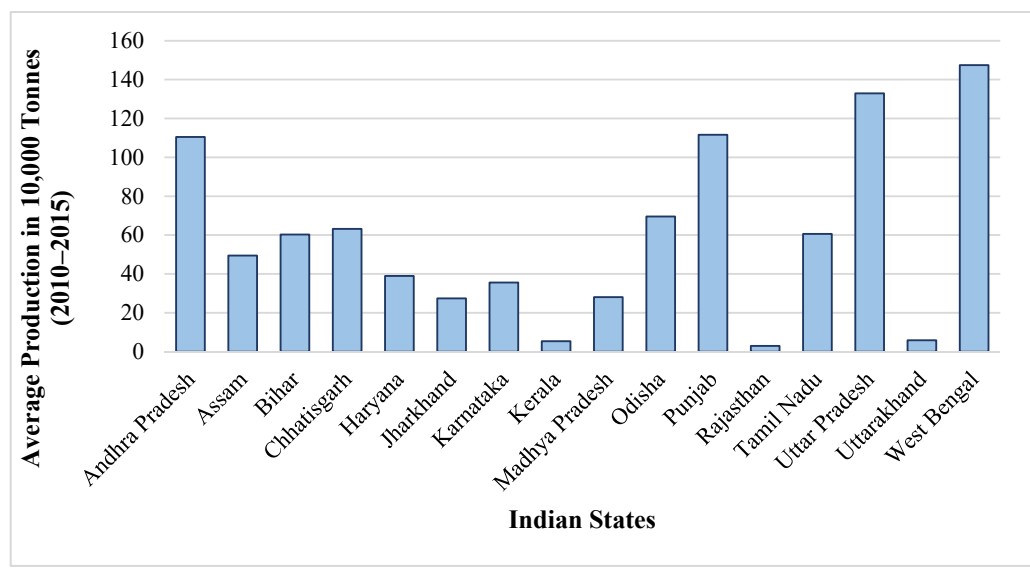

**Figure 2.** State-wise average rice production in India, 2010–2015. Source: Author's elaborations using data from the Directorate of Economics and Statistics, India.

### 1.2. Flood-Tolerant Seed Variety: Swarna-Sub1

Quality seeds are the most basic and crucial input that is required in agriculture production, as the response of other inputs of production is dependent on the quality of seeds being used. Several poor farmers in the flood prone regions are switching from high-yielding varieties of seeds to traditional and local varieties which can withstand submergence to cope with flooding. However, studies indicate that these local and traditional varieties often give very low yield, making it unprofitable for the farmers [17]. With floods affecting every year, in some places, farmers often abandon cultivation and leave their fields fallow during the monsoon season [18]. In areas where high-yielding but submergence-intolerant rice varieties are cultivated, farmers suffer from heavy crop losses caused by recurrent flooding. Amongst recent strategies adopted to overcome the problems

of flooding in agricultural areas is the development and dissemination of high-yielding varieties that are flood-tolerant along with acceptable agronomic and quality traits [13].

The work on the development of flood-tolerant rice varieties was started in the year 1987 at International Rice Research Institute (IRRI), and the submergence-tolerant gene-SUB1A- was developed [18,19]. Since then, several new rice varieties have been developed by introgressing SUB1 gene into high yielding and high yielding rice varieties [13]. The Sub1 gene was fused into several other popular varieties of rice grown in South and South-East Asia, such as *Swarna, Sambha mahsuri, BR11*, etc., which can ensure rice production in flood-prone areas [13,17,20]. Amongst these new varieties, Swarna-Sub 1 (SS1)–a submergence-tolerant rice variety–is considered extremely viable in the flood affected regions in India and has been distributed to rice farmers in eastern India since 2008 [21]. SS1 survives full submergence for up to 14 days as it was developed by introgressing a single quantitative trait locus that causes submergence tolerance in Swarna, which is a popular rice variety in eastern India [19]. Even under normal conditions, SS1 is considered to show no significant differences in agronomic performance, grain yield or grain quality as compared to Swarna [7,21]. Moreover, it is already available in the markets for commercial cultivation. This new variety can ensure rice production in flood-prone areas owing to its tolerance to submergence.

Several studies have documented the performance of SS1 amongst farmers in both experimental and non-experimental settings to understand the seed's yield advantages under different submergence stages. In a non-experimental setting, a study on the farmers in eastern Uttar Pradesh and Odisha on rice production in 2011 *Kharif* season reveals a yield advantage of SS1 compared to Swarna under medium-duration submergence i.e., 8 to 14 days of submergence [19]. In a comparative study by [7] for all Sub1 varieties, including SS1, the study shows that all submergence-tolerant seed varieties were promising, with either similar or higher yield than their non-Sub1 counterparts. Conducting a randomized control trial in flood affected fields of Orissa, ref. [21] found that SS1 had a positive impact on the yield when fields were flooded compared to Swarna. By effectively replacing Swarna with SS1 seeds, a significant improvement in rice production is expected of approximately 9 to 12% [21]. Furthermore, a wide scale adoption of SS1, prior to floods would have resulted in an approximate increase in rice production by 26% [21]. Apart from flood tolerance, the SS1 seeds also reduce the risk of yield loss during the normal growing period i.e., when there is no flood impact [20,22,23]. Overall, the development of submergence-tolerant varieties allows farming communities to become more resilient to existing and growing flooding risks [24].

To cope with recurring flood impacts, flood-tolerant seeds are increasingly being adopted by farmers to bring productivity gains to flood prone areas. Potential benefits of adopting SS1 seed varieties have been explored across India and it has been estimated that a large-scale development and dissemination of SS1 seed varieties can be beneficial for 30–40% or 12–14 million ha of the 44 million ha of rice cultivated area which is exposed to recurrent flooding in the Gangetic basin in India alone [21]. In such conditions, seeds that can withstand flood submergence for a longer period can indeed be a game changer in making small holder farmers resilient to frequent floods. Technology such as Swarna-Sub1 are already available for use [22], however in regions where flooding is predictable, there is a requirement to do a needs assessment for such stress-tolerant seeds and arrange for seed procurement accordingly. Therefore, it is imperative to identify the quantity of certain important seed varieties such as SS1, which can be beneficial in those agriculture areas. Considering the potential benefits offered by the SS1 seed variety in terms of coping with floods, an assessment of the amount of SS1 seed requirement will allow the governments to ensure seed security and cater to the flood-affected farmers need. As per our knowledge there are no systematic in-depth within-country assessments of seed requirement for flood risk management.

Following from the above discussion, this paper presents an assessment of the required flood-tolerant rice seed variety called Swarna-Sub1 (SS1), as an adaptive flood risk

management method during the main rice cropping season called Kharif (or monsoon) at the district level in India. Using a combination of flood area estimates derived from remote sensing data from the period 2000–2018 and land-use data from the government database, this paper provides an estimate of SS1 seeds that would be required as an adaptive flood strategy in seed banks for 17 major states in India. Specifically, the paper presents an estimate of the amount of SS1 seeds that will be required to be maintained in the seed banks and the cost implication on the exchequer for procuring the seeds. Furthermore, these estimates can also be useful for planning in-season flood risk management through the revival of crop production and in maintaining food supply. The quantity of seeds required across different districts show large differences based on the cultivated area and the severity of floods in that district. The estimates from this study provides valuable information to the policy makers, who can make informed investment decisions to establish new seed banks in locations where floods are recurring with a high probability or store additional seeds in existing seed banks. In addition, our analysis shows how remote sensing data can be used in complement with land use data to obtain reliable estimates of seed requirement needs that can improve the preparedness of government departments in procuring necessary stress-tolerant seeds in areas with the most urgent demand.

## 2. Data and Methodology

### 2.1. Mapping Flood Extent Using Satellite Data

To map the extent of long-term flood records the study employed NASA MODIS (MOD09A1) eight-day composite surface reflectance product with 500 m spatial resolution between 2001 and 2018. Furthermore, to determine the severity and the duration of the flooding, two major water indices, namely, Enhanced Vegetation Index (EVI) and Land Surface Water Index (LSWI), were applied to differentiate land and water pixels using the threshold approach [25,26]. In the case of EVI, a threshold value of less than or equal to 0.05 and an LSWI less than or equal to 0, as the first criteria, were adapted. The second criteria applies if the EVI value less than or equal to 0.3 and the difference in the value of the EVI and LSWI (DVEL) values is less than or equal to 0.05, to estimate the overall inundation extent on each 8 day MODIS product. Steps involved in image processing and computation of land and water indices and its thresholds can be referred here [27].

Figure 3 shows the comparison of MODIS Terra satellite data with flood inundation extent for the 2010 flood event clearly shows good agreement with the satellite data. Using ArcGIS Spatial Analyst toolbox, the time series inundation product was produced monthly and annually for the flood extent information, to identify the flood duration and its occurrences over 18 years (2001–2018).

### 2.2. Data for Estimating Seed Requirement

#### 2.2.1. Land-Use Data

Focusing on the cultivation of rice during the monsoon season, the district-wise data on land use was collected from Directorate of Economics and Statistics, Ministry of Agriculture and Family Welfare, Government of India from the year 2000 to 2014. From this, data on the area under paddy sown during the *Kharif* season and the net area sown under all crops was collected. Net area sown, defined as the total area which is sowed at least once in the same year, was used for the analysis. The dataset provides details on the area sown with rice crop by seasons, which are broadly categorised as autumn, winter and summer. The main rice growing season in India is the *Kharif* season and roughly 84% of the rice is grown during this season. The sowing time of winter (*Kharif*) rice is in June–July which is when monsoon arrives and it is harvested between the months of November–January). Given that the focus of this study is on the *Kharif* season, when most of the flooding happens in India, the land-use data on area during *Kharif* season was used.

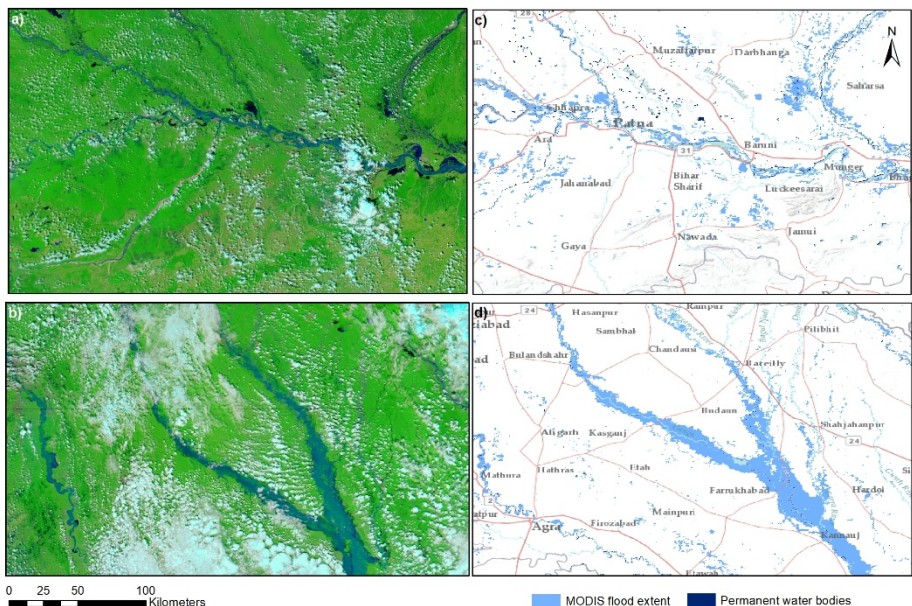

**Figure 3.** (**a**,**b**) MODIS TERRA satellite data taken on 22 September 2010; (**c**,**d**) are the flood inundation extent covering the states of Uttar Pradesh and Bihar, India. Source: IWMI.

Furthermore, the study considers only 17 states for analysis, which are Andhra Pradesh, Assam, Bihar, Chhattisgarh, Haryana, Jharkhand, Karnataka, Kerala, Madhya Pradesh, Meghalaya, Odisha, Punjab, Rajasthan, Tamil Nadu, Uttar Pradesh, Uttarakhand and West Bengal. These 17 states represent approximately 89% of the rice-growing area across the Indian states. The states of Maharashtra and Gujarat are considerably big, both in terms of their economic capacity and agricultural share; however, rice is not the primary grown crop in these regions owing to their agro-climatic conditions which are better suited for growing cotton, sugar cane, millet and so on. Overall, the rice growing area is 5% for both the states combined. The remaining states that were not included in the study were excluded either because district level land use data was not available, for states such as Gujarat, Maharashtra, Manipur and Telangana; or because average area used for growing rice was negligible, such as for the states of Arunachal Pradesh, Himachal Pradesh, Jammu and Kashmir.

2.2.2. Data on Seed Rates

Along with the area under paddy, our estimates will also require the seed rates for paddy cultivation. Generally, the seed rate is understood as the quantity of seed that is required to sow a unit area of land for optimum crop-production. The seed rates vary with the methods of crop establishment by farmers. In India, there are three primary ways of establishing rice for cultivation; *dry seeding, wet seeding* and *transplanting* [28,29]. In the dry seeding method, the dry seeds are sown by either broadcasting, i.e., scattered across the field; drilling, wherein the seeds are drilled manually or mechanically into the field; or dibbling, which is practiced along the mountain slopes using traditional methods and tools to sow the seeds [28]. In the wet-seeding method, pre-germinated seeds are sown in wet puddled soils using the same techniques as broadcasting or drilling. Both the dry- and wet-seeding methods are collectively known as the *direct seeding method*, as the seeds are sown directly using broadcasting, drilling and dibbling [28].

In the transplanting method, the seedlings are first grown in nurseries and later replanted to the main fields, which requires a lot of labour input. One of the reasons for employing such labour-intensive methods for rice cultivation is due to a higher labor supply resulting from population growth [30]. Alternatively, the incentives of adopting the direct seeding method increase when both labour and water availability is low, it and is especially adopted during the dry seasons or in dry regions. Given the higher proportion of

rain-fed rice area in India, it has subsequently led to a lesser adoption of the direct seeding method in India [28,30,31]. Consequently, transplanting became a dominant method for rice establishment in India. The estimates of the percentage of total rice area established by direct seeding method reveals that only 28% of the total paddy area in India is established using this method [28]. Due to unavailability of data on the percentage of paddy area under different sowing methods, this paper assumes that 72% of total rice area in India is established using the transplanting method and 28% of the total rice area is established using the direct-seeding method, following from the above discussion. This is an important assumption in this study, as the seed rates for both the methods vary significantly.

Based on the above discussion, the data on seed rates for each state were collected from the reports published by the state department of agriculture, which includes the details of rice production and output. The data on seed rates based on the method of direct seeding/transplanting are presented for each state in Table 1 (below). For direct seeding, wherever data was available for broadcasting, dibbling or more, an average was calculated. It should be noted that, due to a lack of data on the seed rates for the states of Uttarakhand, Meghalaya and Rajasthan, the data for their neighbouring states were used. For instance, for Uttarakhand, the seed rates used for Uttar Pradesh were used; for Meghalaya, the seed rates of Assam were used; and for Rajasthan, only for direct sowing method, seed rates were unavailable and therefore the seed rates used in Gujarat were used. The final estimation for seed rates is done by taking weighted averaged of the two rice establishment methods, i.e., 28% is cultivated through direct seeding and 72% through transplanting. For example, (see Table 1): for the state of Madhya Pradesh the weighted average will be equivalent to: $(80 \times 0.28) + (50 \times 0.72) = 61.5$, where 0.28 is the percentage of direct seeded rice establishment area and 0.72 is the percentage of transplanted rice establishment area. This was done for all the states and the results are presented in Table 1 (column 3).

**Table 1.** Seed rates for different rice establishment method.

| Sr. No. | States | Seed Rate Using Direct Sowing Method | Seed Rate Using Transplantation | Weighted Average |
|---|---|---|---|---|
| 1 | Madhya Pradesh | 80 | 50 | 58.4 |
| 2 | Andhra Pradesh | 70 | 50 | 55.6 |
| 3 | West Bengal | 70 | 43.33 | 50.8 |
| 4 | Odisha | 80 | 30 | 44 |
| 5 | Bihar | 90 | 30 | 46.8 |
| 6 | Punjab | 17.5 | 25 | 22.9 |
| 7 | Uttar Pradesh | 75 | 21 | 36.12 |
| 8 | Karnataka | 80 | 62 | 67.04 |
| 9 | Kerala | 80 | 60 | 65.6 |
| 10 | Tamil Nadu | 80 | 43.33 | 53.59 |
| 11 | Haryana | 100 | 35 | 53.2 |
| 12 | Chhattisgarh | 80 | 30 | 44 |
| 13 | Jharkhand | 80 | 40 | 51.2 |
| 14 | Assam | 75 | 40 | 49.8 |
| 15 | Uttarakhand | 75 | 21 | 36.12 |
| 16 | Himachal | 80 | 30 | 44 |
| 17 | Rajasthan | 50 | 25 | 32 |

*2.3. Estimation of Seed Requirement*

Using the satellite-derived agriculture mask data from National Remote Sensing Centre (NSRC), India and data on net area sown for rice during the *Kharif* season collected from Directorate of Economics and Statistics, Ministry of Agriculture and Family Welfare, Government of India, the following estimation method was developed to derive the seed requirement for the districts of 17 Indian states.

The estimation of flood-affected *Kharif* paddy area was performed using the combined data on flood inundation and land-use data. Firstly, using the district-level land-use data from 2000–2014, an average percentage of the *Kharif*-paddy sown area as a percentage total net sown area was estimated. From the time series data on flood inundation, the average agricultural flooded area from the year 2001 to 2018 was estimated. Finally, assuming the proportion of flood affected area that is under paddy to be the same as the proportion of net sown area under paddy, we derive the extent of flood-affected *Kharif*-paddy area in each district. From these district-wise estimations of the flood-affected paddy area and using the average seed rates in each state, we estimate the requirement for flood-tolerant seeds that will be required if we want to provide entire flood-affected paddy areas with Swarna Sub1. Further the estimations were also extended to calculate the cost for procuring the required amount of SS1 seeds at the rate of 40 INR per kg (this is the average cost of procuring SS1 seeds in India).

**3. Results**

*3.1. Flood Inundation Mapping*

In this study, the spatio-temporal extent of flooding was assessed for all of India using MODIS TERRA satellite data, which is shown in Figure 4. The map shows the extent of flood severity and its likely impacts on people and agriculture given the large stretch of area is highly prone to flooding and several states are frequently affected due to trans-boundary floods. The flood recurrent map of India shows two main hotspots as the Ganges and the Brahmaputra. These rivers' major tributaries flowing across the states of Bihar, Uttar Pradesh, Assam and West Bengal cause heavy flooding during the monsoon season. As noted in earlier sections, these states are amongst the most flood-prone states in India and are annually affected by flood impacts. Furthermore, from the spatial assessment, weekly flood inundation maps were aggregated into monthly and annual flood inundation maps to derive the frequency of flooding in each pixel across India for a period of 18 years (2001–2018) to produce a flood recurrence map (Figure 4). These flood hotspots quantify the frequent occurrence of flood events in a 20-year mapping period.

*3.2. Seed Requirement Estimations*

Following the methodology described in the previous section, the seed requirement was estimated at district and state level. The state-wise estimates of total seed requirement, total cost of procuring the seeds (at the rate of 40 Rs/kg) and total cost in both Indian Rupee (INR) and US Dollars (USD (1 USD = 70 Rs)) is provided in Table 2.

Overall, our estimates show that 92,764 tonnes of Swarna Sub1 seeds is required for the flood affected paddy areas in 17 Indian states. The cost of procuring the seeds would be approximately USD 53.01 million. Given that Bihar is the most flood-affected state, the seed requirement is also highest for Bihar, i.e., 21,888 tonnes, and the cost of procuring this is estimated at USD 12.51 million. It is estimated that 18,684 tonnes of seeds, costing USD 10.68 million, would be required. Figure 5 presents an all-India map depicting state-wise seed requirements for better illustration.

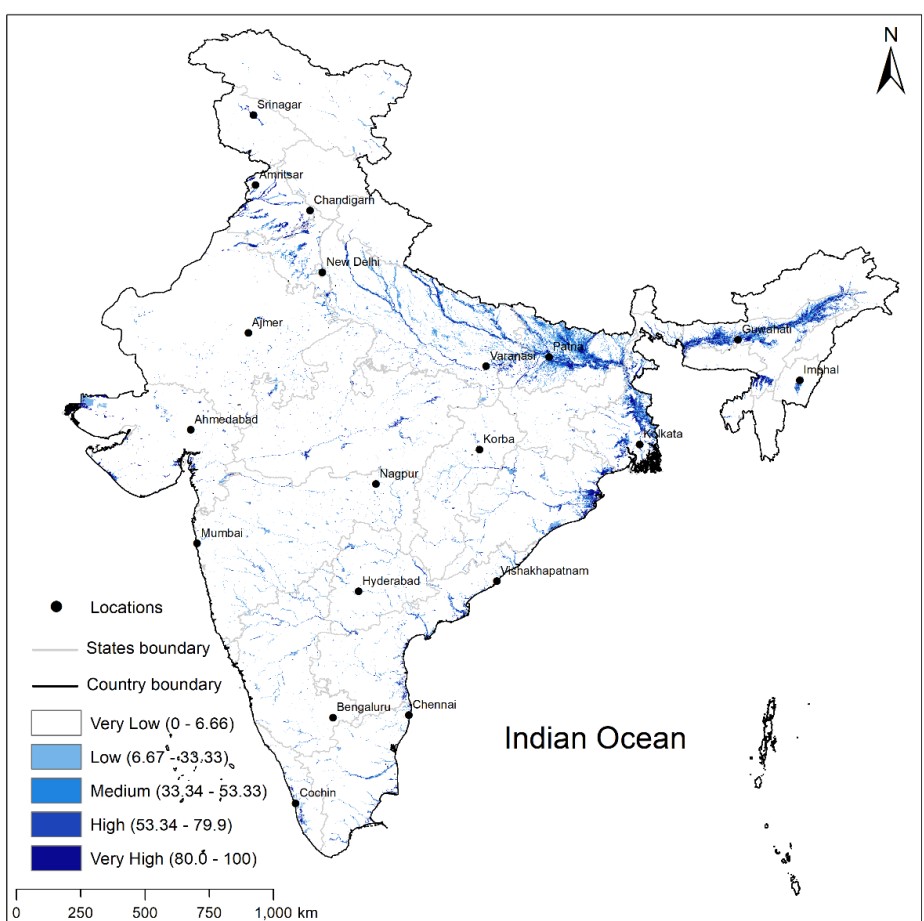

**Figure 4.** Recurrent flooded areas mapped using MODIS satellite data for India.

**Table 2.** State-wise total Swarna-Sub 1 seed requirement and cost of procuring in INR and USD.

| State | Flood Affected Agricultural Area (ha) | Flood Affected Paddy Area (ha) | Seed Requirement (Tonnes) | Cost in Million INR | Cost in Millions USD | Ranking of States Based on Seed Requirement |
|---|---|---|---|---|---|---|
| Andhra Pradesh | 180,152.63 | 58,571.68 | 3256 | 130.24 | 1.86 | 8 |
| Assam | 484,214.47 | 305,204.90 | 15,200 | 608 | 8.69 | 3 |
| Bihar | 1,063,189.47 | 467,756.26 | 21,888 | 875.52 | 12.51 | 1 |
| Chhattisgarh | 31,644.74 | 26,402.19 | 1162 | 46.48 | 0.66 | 11 |
| Haryana | 183,084.21 | 65,467.11 | 3478 | 139.12 | 1.99 | 7 |
| Jharkhand | 28,696.05 | 21,289.74 | 1092 | 43.68 | 0.62 | 12 |
| Kerala | 19,019.74 | 1796.07 | 116 | 4.64 | 0.07 | 14 |
| Madhya Pradesh | 91,780.26 | 20,983.37 | 1224 | 48.96 | 0.70 | 10 |
| Meghalaya | 6477.63 | 1449.99 | 70 | 2.80 | 0.04 | 15 |
| Odisha | 291,390.79 | 248,963.81 | 10,958 | 438.32 | 6.26 | 4 |
| Punjab | 298,363.16 | 211,419.15 | 4844 | 193.76 | 2.77 | 6 |
| Rajasthan | 68,869.74 | 885.49 | 28 | 1.12 | 0.02 | 16 |
| Tamil Nadu | 85,164.47 | 37,249.61 | 1996 | 79.84 | 1.14 | 9 |
| Uttar Pradesh | 486,614.30 | 227,700.22 | 8216 | 328.64 | 4.69 | 5 |
| Uttarakhand | 2217.11 | 605.42 | 22 | 0.88 | 0.01 | 17 |
| West Bengal | 542,385.73 | 367,816.48 | 18,684 | 747.36 | 10.68 | 2 |
| Karnataka | 76,267.11 | 7914.67 | 530 | 21.20 | 0.30 | 13 |
| **Total** | **3,939,531.60** | **2,071,476.16** | **92,764** | **3,710.56** | **53.01** | |

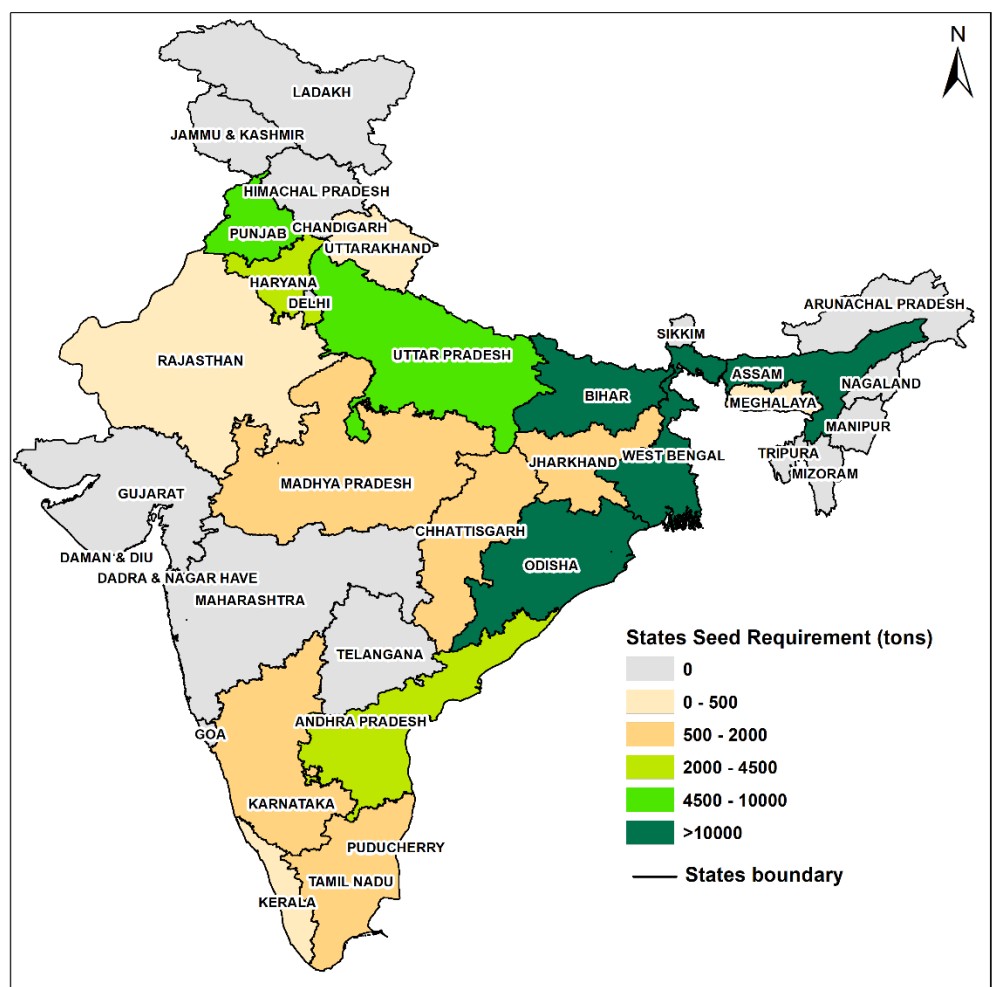

**Figure 5.** State-wise requirement of Swarna-Sub1 seeds across the study area.

The following figure (Figure 6) provides a district-wise map of India illustrating the seed requirements (in tonnes) for each district. At an aggregate level, the five most flood-affected states requiring highest number of seeds are *Bihar, West Bengal, Assam, Odisha* and *Uttar Pradesh.*

Interestingly, it can be seen that while West Bengal is the largest producer of rice (refer Figure 2), Bihar requires the highest amount of seed for adaption. This is because the area in the *Kharif* rice area affected by flooding is higher in the state of Bihar. Delving further into the state-wise requirement, the results appear consistent with the regional variation in flood impacts. For instance, in the state of Bihar, districts like Gaya and Jamui are drought-prone and the estimates reveal a lower seed requirement for these districts. On the other hand, the districts of Patna, Muzaffarpur and Madhubani in Bihar require the highest quantity of seeds owing to large paddy areas affected by flood. In West Bengal, most districts reveal a high seed requirement given that it is the largest rice producer in the country. Only for the district of Darjeeling, which is at a higher altitude (around 2000 m above sea level), is the seed requirement negligible. In Uttar Pradesh, the districts of Siddharth Nagar and Gorakhpur, situated on the eastern part of the state, show the highest seed requirement. In Odisha, the Kendrapara district requires the highest amount of seeds which is approximately 2574 tonnes. In Assam, Cachar and Lakhimpur top the list of districts requiring seeds to cope with floods. Amongst the Southern states, Andhra Pradesh, which is a top rice-producing state, requires an overall 3256 tonnes of SS1 for its flood affected farms. For the districts of Karnataka and Kerala, the requirement of SS1 is comparatively low. Moreover, in the northern and norther-eastern states such as Haryana,

Meghalaya and Uttarakhand, the seed requirement for some districts is zero, which is most likely because these districts are either not affected by floods or they do not cultivate rice.

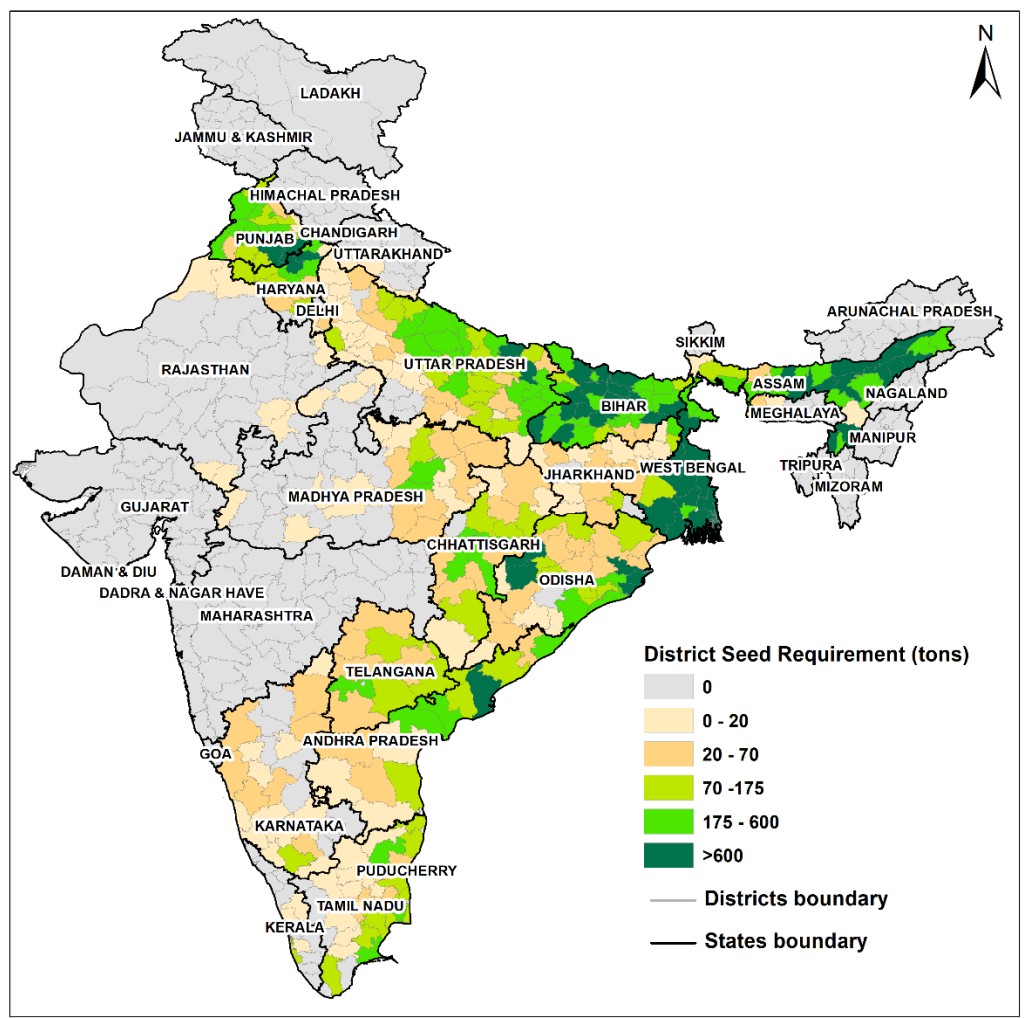

**Figure 6.** District-wise seed requirement estimation for Swarna-Sub 1 for the selected states of India.

## 4. Discussion

Despite growing economically, Indians are still hugely dependent on agriculture for their livelihoods. Floods have been a major natural disaster (along with droughts) which impacts Indian agriculture sector heavily. Between 1950–2020, India had 304 major flood events, which affected around 895 million people in the country and cost USD 84 billion [1]. With exacerbating changes in global climate, the severity and intensity of floods will likely increase further. Therefore, there is an urgent need to find ways to adapt to floods and secure poor and smallholder farmers' livelihood. With technological advancement, scientists have developed several viable and stress-tolerant seed varieties which can withstand disasters such as floods and droughts. Given the availability of such seed technologies, there is still a need to understand the amount of seeds required by farmers. With this regard, this study explored the amount of flood-tolerant seeds of the *Swarna-Sub 1* rice variety that would be required by rice cultivators (during the monsoon season) at the district level of India's 17 major states. The estimates reveal an overall requirement of 92,764 tonnes of SS1 rice variety for the flood-affected *Kharif*-paddy areas across all the states. In monetary terms, this would cost approximately 370 crore INR (or USD 53 million) to the exchequer. Considering the government's total spending on the agriculture sector, USD 53 million amounts to less than 0.01 percent of the Indian central government's budget allocation to the Ministry of Agriculture and Farmers' Welfare in 2020. These estimates can

be benefited by potential private sector investors at the regional, national and international level, and stakeholders in the seed sector, while making decisions focusing on flood-risk management in agriculture. While great progress has been made in developing advanced flood-tolerant seed varieties, the question remains if these estimates are viable considering if the government or private sector can establish a seed distribution system to ensure all the flood-affected farmers benefit from this.

Focusing on climate change adaptation, an immediate focus should be given to the agriculture areas which are severely affected by floods annually. As noted earlier, around 12 to 14 million ha of rain-fed rice cultivation area could be benefited if the SS1 variety was adopted by the farmers, as it offers better yield even during flood events [21]. With regards to SS1's adoption and awareness amongst the farmers in India, it can be noted that these seeds have been distributed to farmers, especially in eastern India, since 2008. Since then, the distribution of SS1 seeds has expanded significantly, in particular when the National Food Security Mission, which is a central government initiative to increase the annual production of rice, wheat and pulses, included these seeds in its eastern India programs in 2010 [19,24]. From 2010–2012, around 38,000 tons of paddy seeds were distributed, which reached 1.3 million farmers in Eastern India [21]. Another such initiative to promote and increase SS1's adoption amongst farmers called the 'Stress Tolerant Rice for Africa and South Asia' (STRASA) was funded by the Bill and Melinda Gates Foundation [21]. These initiatives are slowly bringing in changes and an increasing demand and adoption for the same can be seen in the future.

However, as estimated in this study, the production and dissemination of approximately 90,000 tons of SS1 seeds is still a major concern. In India, the production of seeds has a well-established channel for production [32]. The production is done on the basis of indents from either the private or public sector organizations placed with state or central government institutions such as the Department of Agriculture Cooperation (DoAC), Government of India or State Agriculture University/ National Seed Producers, who consolidate the indents and forwards them to Indian Council of Agriculture Research (ICAR) [33]. These seeds are then supplied to indenting organizations on the basis of allocation by DoAC for multiplication of seeds which are later made available to the farmers. This entire process takes at least 3 years, which implies that the public sector seed companies/state governments should have a pre-determined requirement of seeds at least 3 years in advance [34]. Assessing the requirement of seeds in case of contingency can provide a way to pre-plan and procure large quantities of seeds. Amongst other ways of making these SS1seeds available at appropriate times and affordable prices is by deploying a suitable model, such as a participatory seed production method involving both farmers and local stakeholders in the process; enabling partnership with private sector; seed village scheme; and creating awareness through self-help groups and community-based organizations [33–35]. This can be achieved through a continuous interaction between various institutions, policymakers and concerned stakeholders, which can further strengthen the existing local seed systems. Moreover, these interactions will enhance seed productivity and availability, thereby enabling the distribution of flood-tolerant seeds to farmers in distress across the regions [33].

Apart from these structural problems in procurement, distribution and dissemination of the seeds, there is little understanding of how the socio-economic characteristics of the farmers play a role in the adoption of this technology. For instance, in the field experiment conducted in Orissa to understand the yield variability of SS1, the authors point that caste, a marker of social status, in India-played an important role in the adoption of stress-tolerant seeds [21]. Firstly, the study found that the plots cultivated by farmers belonging to marginalized caste groups were already exposed to more flooding. Despite being more vulnerable to flooding, the study found a lower adoption rate of the SS1 seeds amongst them, which was attributed to the high incidence of poverty amongst the lower-caste groups. Thus, there are substantial social barriers in effective adoption of new

technology such as SS1 amongst farmers. This will require designing policies that can make stress-tolerant seeds affordable for poor farmers through subsidies or other benefits.

## 5. Conclusions

The incidences of flood have increased over the years owing to global climate change. Enhancing livelihood options for the people depending on agriculture in several flood affected regions in India is challenging. Despite the technological improvements in terms of development of new tolerant seed varieties, or techniques, poor farmers often adapt to these flood impacts by abandoning farming altogether. To enhance the capacity of the farmers to deal with extreme flood events, advanced flood-tolerant seed varieties can be adopted. To this end, this study provides estimation of one such flood-tolerant rice variety–Swarna-Sub 1 for rice–which can be adopted by farmers during the major rice growing season called *Kharif* in India, at the district level for 17 major Indian states. The total seed requirement for the 17 states is approximately 92,800 tonnes, and the cost for procuring this amount was estimated to be INR 3800 million.

However, our estimates are the first attempt (as far as our knowledge) to quantify the potential need for flood-tolerant seeds such as Swarna-Sub1 that would need to be procured if major flood prone paddy/rice areas in India had to be supplied with flood-tolerant seeds. Our methodology of combining remote sensing data for flood affected areas with land use pattern data can be used for need assessment by government departments to be better prepared in procuring seeds and allocating budget in enhancing agriculture resilience and flood proofing the vulnerable smallholder farmers. Future work would necessitate collecting more data on seed rate usage and sowing methods that can facilitate more accurate estimates. Moreover, future work can also explore the requirement for other major food grains along with considering the impact of droughts.

The procurement of stress-tolerant seeds is not enough by itself to ensure adoption by farmers, who will need these stress-tolerant seeds to become climate resilient. There is need to design integrated flood management policy to attract more farmers in these floods affected areas to adopt stress-tolerant seeds such as Swarna Sub1, and promote the use of climate information services and good agronomic practices in reducing crop losses. Although national and international agencies are beginning to recognize the extent to which extreme weather events such as flooding will affect agricultural production in India, their initiatives to adapt and cope with floods have been relief-oriented and rather short term [36]. Given the increasing flood occurrences, there is a need to develop a comprehensive seed production and dissemination strategy for rapid and targeted distribution of this flood-tolerant rice variety, Swarna-sub1, especially amongst the poor smallholder farmers residing in the flood-prone regions of India. Additionally, workshops and awareness drives on specific climate resilient technologies suitable for that region need to be undertaken to enable farmers to cope with extreme weather events and further enhance their adaptive capacity. In addition, training and capacity-building programs are necessary to enable farmers to adopt best practices and resilient technologies to increase yields.

**Author Contributions:** G.A. and N.K. developed the ideas for the project, analyzed data, interpreted results, and wrote most of the manuscript. N.K. also spent considerable time in data collection. All authors have read and agreed to the published version of the manuscript.

**Funding:** This research was funded by the CGIAR Research Program (CRP) on Climate Change, Agriculture and Food Security (CCAFS), and CGIAR Research Program (CRP) on Water, Land and Ecosystems (WLE), which is carried out with support from the CGIAR Trust Fund and through bilateral funding agreements. For details, please visit https://ccafs.cgiar.org/donors and https://wle.cgiar.org/donors.

**Institutional Review Board Statement:** Not applicable.

**Informed Consent Statement:** Not applicable.

**Data Availability Statement:** The data presented in this study are available on request from the corresponding author.

**Acknowledgments:** The authors would like to thank the Indian Council of Agricultural Research (ICAR) and Japan's Ministry of Agriculture, Forestry, and Fisheries (MAFF). Further, the authors would also like to thank Archisman Mitra and Alok Sikka for their support in conducting the research and providing feedback. We thank Niranga Alahacoon, IWMI in producing GIS maps. Authors would like to thank the reviewers who helped in improving the manuscript.

**Conflicts of Interest:** The authors declare no conflict of interest.

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
