# Peer review of "Geospatial Assessment of Flood-Tolerant Rice Varieties to Guide Climate Adaptation Strategies in India"

_climate, doi:10.3390/cli9100151_

Round 1

Reviewer 1 Report

Dear authors, check specific comments on English sentence structure, Grammar error, and Figures mistakes throughout the manuscript and suggested modified changes for the improvement of the manuscript

  • Line 1: author suggested structuring the article as per MDPI format.
  • Line 2: Suggest changing the title for achieving the objective and show interest for the reader.
  • Line 9: Write statement regarding Rice Crop under flooding condition under abstract section.
  • Line 8: The author should add a statement for future research work in the abstract section
  • Line 8-17: Write the specific objective of your study in the abstract.
  • Line 18: Write some suitable keywords e.g., techniques of Geospatial/Remote Sensing which are adopted in this study.
  • Line 97-101: There is no need to describe the structure of the research article, should be deleted.
  • Line 67-68: If this statement is an assumption of the author, so what is the basis.
  • Line 69: “Technology like Swarna Sub1 are already available for use” should be cited.
  • Line 102: There is no need for background, I suggest adding in the introduction section and concise the paragraphs.
  • Line 209: Author should mention the reason for not using Sentinel and Landsat data for mapping flood.
  • Line 218: Write full form of DVEL.
  • Line 219: ArcGIS is a software package so the author should write which techniques are used for processing.
  • Line 226: why only data collection for the Year 2000 to 2014 and what is the period of your study?
  • Line 232: Check sentence what is mean by (DR.
  • Line 333: Write the date of the satellite image which are used in Figure 4 and indicate A and B for figures under combination, also the Bihar image is not the same satellite imagery.
  • Line 354: Correction in Legend for “state boundary” the layer is not the part of Seed requirement so it must be mentioned separately in Figure 5.
  • Line 359: Should correct the title of Figure 6 as across the study area not India and also some correction for the legend as indicated in the previous comment.
  • Line 380: Concise the discussion section and indicated the most important results.
  • Line 462: Should check the grammar and structure of the paragraph.
  • Line 476: Author should rewrite the conclusion in a technical way because this showing the description of the introduction which is not meaningful for the readers.
  • Line 515: Should write a statement regarding future research work based on your study.

Author Response

Dear Editors

We are sharing here with you the reviewers response for your kind consideration. We have carefully reviewed all the comments shared by the reviewer and revised in the manuscript.

Regards

Giriraj 

Reviewer 2 Report

The authors analyzed the “Geospatial Assessment of Flood-Tolerant Rice Varieties for adaptation to a Changing Climate in India” and I found it satisfactory to publish in CLIMATE. The manuscript is well prepared. In general, I have some technical comments that the Authors should take into account. The Introduction; Results and Discussion section demand further improvements. Moreover, reference must be organized according to the MDPI guidelines.

Abstract need to be revised more descriptive to the point (including the research aim, objectives of your project, and the analytical methodologies applied)

-line 14 Major states based on?? Population or rice production?

-line 16 States or districts??

In literature review author must consider the recently published articles on adaptation to a Changing Climate in other countries as following:

  1. Quantifying climate‑induced drought risk to livelihood and mitigation actions in Balochistan. Journal of Natural Hazards. https://doi.org/10.1007/s11069-021-04913-4
  2. Adaptation and Management Strategies of Wheat (Triticum aestivum L.) Against Salinity Stress to Increase Yield and Quality. Front. Agro. Vol. (3) 661932 (https://doi.org/10.3389/fagro.2021.661932 )

Figure 2 caption detail is not clear. Remake this figure

Figutre 3 Is it a derived product from MODIS? Because the processing how flood map is created is not clear in the methodology.

Figure 4 What are we trying to communicate from the figure is not clear. In the first one label should be Bihar?

More explanation needed on how this flood maps are created?

Figure 5 Use correct India shape file! Uttaranchal is changed to Uttarakhand.

Figure 6 Same comment as Figure 5

Line 508 Use consistent citation.

Author Response

(The authors gave the same response as above.)

Round 2

Reviewer 1 Report

Congrats for having properly implemented the required revisions. I consider the paper publishable.

 Dear authors, I really appreciate your efforts in improving your manuscript, but, in my opinion, it is necessary to incorporate again the same comments and suggestions, to make this work suitable for publication.